# Entropy-Inspired Aperture Optimization in Fourier Optics

**DOI:** 10.3390/e27070730

**Published:** 2025-07-07

**Authors:** Marcos Miotti, Daniel Varela Magalhães

**Affiliations:** Sao Carlos Institute of Physics, University of Sao Paulo, IFSC–USP, Sao Carlos 13566-590, Brazil

**Keywords:** Fourier optics, optical imaging, applied information theory

## Abstract

The trade-off between resolution and contrast is a transcendental problem in optical imaging, spanning from artistic photography to technoscientific applications. To the latter, Fourier-optics-based filters, such as the 4*f* system, are well-known for their image-enhancement properties, removing high spatial frequencies from an optically Fourier-transformed light signal through simple aperture adjustment. Nonetheless, assessing the contrast–resolution balance in optical imaging remains a challenging task, often requiring complex mathematical treatment and controlled laboratory conditions to match theoretical predictions. With that in mind, we propose a simple yet robust analytical technique to determine the optimal aperture in a 4*f* imaging system for static and quasi-static objects. Our technique employs the mathematical formalism of the *H*-theorem, enabling us to directly access the information of an imaged object. By varying the aperture at the Fourier plane of the 4*f* system, we have empirically found an optimal aperture region where the imaging entropy is maximum, given that the object is fitted to the imaged area. At that region, the image is lit and well-resolved, and no further aperture decrease improves that, as information of the whole assembly (object plus imaging system) is maximum. With that analysis, we have also been able to investigate how the imperfections in an object affect the entropy during its imaging. Despite its simplicity, our technique is generally applicable and passable for automation, making it interesting for many imaging-based optical devices.

## 1. Introduction

When it comes to imaging solutions, the balance between contrast and resolution is not easily achieved. In trendy fields such as super-resolution optical microscopy [1], high-resolution imaging of highly scattering media [2], and noninvasive techniques for biomedical applications [3], it is difficult to find a general configuration in which brightness and detail solving are mutually compensated given an imaged object. Therefore, different methodologies are found in the literature for the same technique in different contexts. Moreover, advancements in artificial intelligence-based digital image processing [4] are shifting the task of handling contrast and resolution from hardware to software, as well as dealing with the modeling and explainability of real-time detections [5,6] in the place of rigid mathematical equations. Under that perspective, we decided to tackle the contrast–resolution balance problem in static and quasi-static imaging for a well-established field: Fourier optics. In particular, we were looking for an empirical approach to analyze the problem purely from the standpoint of the *information carried by an image*, which led us to entropy.

In terms of image enhancement, Fourier optics have been known for decades [7] for their filtering properties, exploiting the optical Fourier transform suffered by light rays crossing a spherically surfaced lens and the removal of high spatial frequencies by adjusting the aperture of an iris diaphragm positioned at the Fourier plane of the lens. More specifically, in 1961, O’Neilley and Asakura [8] presented a formal theory connecting the information of a Fourier-transformed optical image with its formation, providing a means to calculate the image-related statistical entropy *H*. In 1998, Kriete [9] connected entropy calculation to image quality in the context of fluorescence microscopy. In the context of non-imaging optics, in 2007, Markvart [10] provided a direct thermodynamic description of the light rays confined in an optical system in terms of their *étendue* or throughput, which is essential for efficient optical design. Furthermore, in the general context of time-dependent signal processing, the entropy of a signal and the entropy of its Fourier transform are inversely related by means of an entropic uncertainty principle [11]. Interestingly enough, few subsequent works [12] have employed consistently entropic methods to assess the information of optical images, which, to us, is a key point for addressing the contrast–resolution balance problem. As a consequence, we found no investigation about mutual contrast–resolution enhancement by optimizing the aperture at the Fourier plane, i.e., the sole degree of freedom in a Fourier-optics-based imaging system, which motivated this study.

Our goal with this paper is two-fold: (1) to present an analytical technique of practical use for experimentalists and (2) to discuss, in a heuristic manner, a different viewpoint on statistical entropy in the context of optical imaging, enabling us to understand an image as simply as the information it carries. In view of that, we discuss the *H*-theorem in Section 2 as a motivation to link entropy and optical imaging in this work. In Section 3, we review the essential points of Fourier optics for understanding our experiment, whose preparation is described in Section 4. Our analytical technique is evaluated in Section 5, with an optimization algorithm for its implementation described in Section 6. An overview of our findings is presented in Section 7.

## 2. Opto-Statistical Insight into the *H*-Theorem

As a motivation to build our intuition connecting entropy and optical imaging, we will quickly discuss the conditions and the result of the *H*-theorem, following Reif’s textbook. Ref. [7] considers an isolated system, defined by a series of approximate quantum states, where the effect of interactions is small. At any time instant *t*, the probability $P_{s}=P_{s}\left(t\right)$ of the system being found in a particular state *s* is governed by the following law:(1)$$\frac{dP_{s}}{dt}=\sum_{s^{\prime }\neq s}W_{s⇌s^{\prime }}\left(P_{s^{\prime }}-P_{s}\right);\;\sum_{s}P_{s}=1,$$
in which $W_{s⇌s^{\prime }}$ is the probability per unit of time that the system makes a transition from state *s* to state $s^{\prime }$ or *vice versa*, a condition known as “detailed balance”. To see what can be found from the dynamics in Equation (Equation 1), the quantity that denotes the theorem is defined as follows:(2)$$H\equiv \underset{s}{\mathrm{mean}}\left(\log(P_{s})\right)=\sum_{s}P_{s}\log(P_{s}).$$

In the context of statistical mechanics, the quantity in Equation (Equation 2) is regarded as the non-equilibrium entropy of the system. Notice that because $0\leq P_{s}\leq 1$, so is H≤0. Furthermore, its time derivative (which can be obtained from Equations (Equation 1) and (Equation 2) after a few algebraic steps) provides the main result of the theorem about the dynamics of the system’s states:(3)$$\frac{dH}{dt}\leq 0\;\Rightarrow \;\frac{dH}{dt}=0\;\mathrm{for}\;P_{s}=\mathrm{constant}\;\forall s.$$

Hence, dH/dt will always be negative, except when all states are equally probable. In that case, the system is said to have reached equilibrium, as the entropy *H* attains its maximum (least negative) value, analogous to the thermodynamic potential of the same name.

As noticed by O’Neilley and Asakura [8], an image’s *H* as a function of the defocusing also satisfies the conditions in Equation (Equation 3), reaching its maximum value at the focus. Therefore, *H* encompasses information about the *quality* of the image somehow, which in turn is tied to both contrast and resolution. By reducing the aperture of an iris placed at the Fourier plane of a lens, one enhances the resolution by filtering higher spatial frequencies while decreasing the contrast for blocking part of the light. As a sharp image balances contrast and resolution, there should be an *optimal* aperture for the best image quality, maximizing the entropy of the image. The details to achieve those conditions will be analyzed in Section 3.

## 3. Fourier Optics and Imaging Entropy

We recall the essential features of Fourier optics for this paper following Chapter 5 in Goodman’s textbook [13]. Consider the plano-convex lens in Figure 1 of focal length *f*, illuminated by a monochromatic plane-wave light source of amplitude $U_{\max}$ and wavelength λ in the geometrical limit. For an object standing one focal length away on the left-hand side of the curved surface, the transversal electromagnetic field $U(x_{f},y_{f})$ of the image formed at the focal point on the right-hand side of the curved surface is the Fourier transform (denoted as F) of the transversal electromagnetic field at the object’s position $U\left(x_{o},y_{o}\right)=A\tau \left(x_{o},y_{o}\right)=A\tau_{o}$ (τ being the lens’ transference function), as shown in Equation (Equation 4):(4)$$U\left(x_{f},y_{f}\right)=\frac{U_{\max}}{𝚤\lambda f}e^{𝚤\frac{\pi }{\lambda f}\left(x_{f}^{2}+y_{f}^{2}\right)}\underset{\mathrm{object}\;\mathrm{plane}}{\;\int \int \;}\tau \left(x,y\right)e^{𝚤\frac{2\pi }{\lambda f}\left(xx_{f}+yy_{f}\right)}dxdy=F\left[U_{o}\right]\left(\frac{x_{f}}{\lambda f},\frac{y_{f}}{\lambda f}\right).$$

Note that the mapping of the incident plane wave into a focusing spherical wave in Figure 1 is reversible if we consider a point-like source at the focal point. Therefore, by mirroring the plano-convex lens into a biconvex lens and placing a copy of it two focal lengths away from the first one, we find a configuration called the 4*f* system, which was used in the experiment described in Section 4. That arrangement allows us to Fourier-transform a signal, filter spatial frequencies $\left(x_{f}/\lambda f,y_{f}/\lambda f\right)$ of the resulting field at the focal point or Fourier plane, and then inverse-Fourier-transform the filtered signal, improving imaging.

As commented before at the end of Section 2, an iris diaphragm (often simply called iris) is placed at the Fourier plane, blocking part of the signal and thus filtering the Fourier-transformed field. The smaller the aperture (also called the pupil, in contrast with the iris that blocks light), the more spatial frequencies will be removed. By modeling the aperture as $A=A(\left(x,y\right),\left(x_{f},y_{f}\right))$, Equation (Equation 4) informs us of the *U*-field at the image plane of a 4f system:(5)$$U_{i}=U\left(-x_{i},-y_{i};A\right)=F^{-1}\left(F\left[U_{o}\right]*F\left[A\right]\right),$$
in which ∗ is the convolution operator. Provided a static or quasi-static object, the only degrees of freedom that change $U_{i}$ come from the aperture function *A*. For a circular-aperture iris, it simplifies to a single degree of freedom, the aperture diameter *a*.

To access the information carried by $U_{i}$, we measure the intensity of the light signal since $I\propto {\left|U\right|}^{2}$. Using that, O’Neilley and Asakura [8] calculated the imaging entropy as follows:(6)$$H=\int_{S_{i}}p\left(I_{i}\right)\log\left(p\left(I_{i}\right)\right)dI_{i};\;\int_{S_{i}}dp=1,$$
in which *p* represents *portions* of the intensity $I_{i}$ sampled across the image plane’s photosensitive surface $S_{i}$. Observe that Equation (Equation 6) is mathematically equivalent to Equations (Equation 1) and (Equation 2). As discussed previously, the degrees of freedom to maximize *H* in the imaging case, thus satisfying the result of the *H*-theorem in Equation (Equation 3), come uniquely from the aperture function *A*. Thus, considering the photosensitive surface to be a pixelated camera sensor, we can rewrite Equation (Equation 6) in the dimensionless summation form:(7)$$H=H\left(A\right)=\sum_{m,n}p\left(I_{m,n}\left(A\right)\right)\log\left(p(I_{m,n}\left(A\right))\right);\;\sum_{m,n}p\left(I_{m,n}\left(A\right)\right)=1,$$
in which (m,n) represents the pixel coordinates, where the imaging intensity $I_{m,n}\propto {\left|U_{m,n}\left(A\right)\right|}^{2}$ will produce a local detection response. The sum of the individual pixel responses (generally a number between 0 and 255) across the array is normalized to one, as represented by the portion function *p*. Of note, the imaging entropy depends explicitly on the aperture function in Equation (Equation 7), which for a circular aperture assumes the following form:(8)$$A=A_{\mathrm{circ}}\left(\rho ,a\right)=\left\{\begin{array}{cc} 1 & \;\mathrm{if}\;\rho \leq a.\\ 0 & \;\mathrm{if}\;\rho >a. \end{array}\right.$$

Hence, we can look for the circumstances for maximizing the entropy as a function of the aperture *a*. An experiment for finding such an optimal condition is the subject of Section 4.

## 4. Experimental Setup

In this section, we discuss how we prepared an experiment to quantify the statistical entropy of an imaged object as a function of the optical aperture. In Section 4.1, we show the 4f imaging system built for that purpose. In Section 4.2, we discuss the handcrafting process to make the absorptive mask used as the imaged object, the so-called *object mask*.

### 4.1. Imaging System

We assembled the imaging system described in Figure 2 in the corner of an optical table, surrounded by a covering structure that creates a dark environment over the table when closed. We set up this experiment for remote operation, as the object mask (OM) remained unchanged, and the motorized iris (MI) had a cable connecting it to the control laptop, which could access the *Raspberry*-controlled camera sensor (CS) through a wireless link.

We set the exposure time of the camera sensor (CS) to 130μs, the shortest value for that model. We also did not lower the 0.95 mW nominal power of the laser source (LS), as we wanted to test the robustness of our analytical technique, possibly having to deal with saturated pixel values. With that in mind, we also designed an object mask (OM) to serve as an imperfect imaged object, projecting its shadow onto the CS, as explained in Section 4.2.

### 4.2. Object Mask

Although originally unintended, the imperfections introduced in the object mask shown in Figure 3 after its handcrafting process led us to retain it to be used in our experiment. After all, measurement artifacts are present in the majority of real-world problems. Therefore, as we desired to test our method under realistic conditions, we proceeded with the measurements using that imaging target.

Now the conditions of the designed experiment are known, we proceed to investigate the measurements obtained from it in Section 5, effectively applying our analytical technique.

## 5. Validating the Technique

From the imaging methods described in Section 4.1, we acquired 360 images of the object mask shown in Section 4.2, with 10 images per aperture value. The acquisition started at a=11.5mm and proceeded in steps of Δa=0.3mm down to a=1.0mm. We also acquired 10 images shot in the dark, thus capturing the background noise. Two examples of the acquired pictures are presented in Figure 4, showcasing the image enhancement capabilities of the 4f system. In Section 5.1, we present how we carried out the post-processing of those images. In Section 5.2, we show how the imaging entropy *H* of each image was obtained, followed by an inspection of the H×a diagrams in Section 5.3, allowing us to find the optimal aperture region. To verify that both contrast and resolution are represented by the imaging entropy and maximized at the optimal aperture region, we assigned metrics to evaluate these parameters in Section 5.4. Finally, we analyze the physical meaning of the optimal aperture and investigate the imaging behavior around the optimal aperture region in Section 5.5.

### 5.1. Image Post-Processing

Each image file is represented as a 3×1944×2592 tensor, corresponding to the three color channels (RGB) and the 1944×2592 pixel arrays captured by the camera sensor for each color. As the laser wavelength was 632.8 nm, we used only R-channel data, converting the images into pixel matrices, whose values range between 0 and 255. We computed the definitive image as the mean (μ-matrix) with its standard deviation (σ-matrix) of the 10 acquired images, resulting in two characteristic matrices for each aperture. For saturated and high-variance (μ+5σ≥255) pixels in the μ-matrix, their values were assigned to −1, excluding them from the calculations in Section 5.2. To distinguish between viable and unviable pixels in the μ-matrices (i.e., the definitive images), we created a custom colormap, setting −1 as white, 0 as black, and 254 as red, as shown in Figure 5. The full list of 36 final images (means plus their standard deviations) is presented in Appendix A.

The sequence of images in Figure 5 displays how the features defining the letter “A” become more distinguishable as the aperture is reduced, making them darker. Moreover, imperfections in the object mask also become more evident at smaller apertures, artifacts that will be considered later in our analysis. Those images will be revised when we analyze the behavior of the entropy, whose calculation is described in Section 5.2.

### 5.2. Entropy Calculation

To calculate the entropy of each image, we adopted the following procedure:Subtract the background noise: $\mu^{\prime }=\mu -\mu_{\mathrm{BG}}$ and $\sigma^{\prime }=\sigma +\sigma_{\mathrm{BG}}$ pixel-wise.Remove negative- and zero-valued pixels from the calculations.Normalize viable pixels by their sum, satisfying the continuity condition in Equation (Equation 1).Calculate the entropy by applying the normalized pixel values ($\tilde{\mu }$) to Equation (Equation 2).

After the steps above, each μ-matrix representing an image effectively becomes a table of portions segmenting the image, as intended in Equation (Equation 6). However, as the normalized pixel values are very small, computing their logarithms can be hard. Thus, we used the following method:Take the absolute order of magnitude of the smallest pixel value: $M=\left|O({\tilde{\mu }}_{\min})\right|$.Find the logarithms as $\log\left(\tilde{\mu }\right)=\log(\tilde{\mu }\cdot 10^{M}/10^{M})=\log\left(\tilde{\mu }\cdot 10^{M}\right)-M\log\left(10\right).$

With the entropy determined, we examine its dependency on the aperture in Section 5.3.

### 5.3. Entropy vs. Aperture Diagrams

To investigate the statistical quality of the collected data, we filtered the pixel values from the μ- and σ-matrices between the procedures from Section 5.1 and Section 5.2 to retain only those satisfying selected thresholds for the coefficient of variation, σ/μ, such that the ratio was less than or equal to the specified values. This allows us to observe how robust the overall behavior of entropy as a function of aperture remains as the coefficient-of-variation threshold for pixels is progressively reduced. Therefore, the H×a diagrams for selected σ/μ curves are shown on a linear scale in Figure 6 and on a log–log scale in Figure 7.

We highlighted in Figure 6 and Figure 7 an orange-shaded region that marks an abrupt change in the behavior of the curves, which we refer to as the “transition region”. For apertures of a≥3.4 mm (above the transition region), the entropy was calculated from all images shown in Figure 5a,b, whereas for apertures of a≤3.1 mm (below the transition region), the entropy was determined from all images shown in Figure 5c,d. We will analyze the regions above and below the transition area in Section 5.4, arguing that the aperture values within the transition region offer a robust compromise between contrast and resolution, significantly improving the imaging conditions.

### 5.4. Contrast and Resolution Metrics

To show that the imaging entropy *H* from Section 5.2 encompasses both contrast and resolution as a global quality metric, we must define metrics to quantify both contrast and resolution individually, observing how they behave as functions of the aperture *a*. For the contrast, one might measure how much the pixel intensities vary within an image; the greater the difference, the greater the contrast. A straightforward metric for that is the *standard deviation of intensity*, defined for a μ-matrix representing an image as follows:(9)$$\mathrm{SDI}\left(\mu \right)=\sqrt{\frac{1}{N}\sum_{m,n}{\left(\mu_{m,n}-\bar{\mu }\right)}^{2}};\;\bar{\mu }=\underset{m,n}{\mathrm{mean}}\left(\mu \right),$$
which is plotted as a function of the aperture in Figure 8.

For the resolution, one could compute how prominent the edges in an image are, which can be carried out by computing the bidirectional gradient of said image. For that task, Sobel operators are well-known tools for edge detection, and we use them to write our resolution metric for an μ-matrix representing an image as its *average squared gradient magnitude*:(10)$$G\left(\mu \right)=\underset{m,n}{\mathrm{mean}}\left({\left(S_{x}*\mu_{m,n}\right)}^{2}+{\left(S_{y}*\mu_{m,n}\right)}^{2}\right);\;S_{x}\equiv \left(\begin{array}{ccc} -1 & 0 & 1\\ -2 & 0 & 2\\ -1 & 0 & 1 \end{array}\right);\;S_{y}\equiv -S_{x}^{⊤}.$$

Again, ∗ denotes the convolution operator, and ${\left\{S_{w}\right\}}_{w=x,y}$ are the *w*-direction 3×3 Sobel kernels; their convolution with the elements of the μ-matrices allows us to determine the gradients in the *x*- and *y*-directions. To handle the discontinuities seen in Figure 5 due to the saturated pixels, we set their values to zero within their normalized μ-matrices. With that, we computed the resolution metric and plotted it as a function of the aperture in Figure 9.

Interestingly, the overall behavior displayed by the data curves in both Figure 8 and Figure 9 is captured by the corresponding entropy curves in Figure 6 and Figure 7. This confirms the use of entropy as an image-quality metric, as initially pointed out by Kriete [9], now linked explicitly to both contrast and resolution. Now that the optimal aperture region is empirically evident, its physical meaning in terms of the system’s operation will be addressed in Section 5.5.

### 5.5. Physical Meaning of the Transition Region

Returning to the discussion in Section 3, we must understand the role of the aperture *a* on the iris and what is quantified with the imaging entropy *H* to understand the results found throughout Section 5. In a 4f system, the iris placed at the Fourier plane acts as a spatial-frequency shutter, blocking optically Fourier-transformed light rays outside the opening of diameter *a*, i.e., the aperture. In terms of the spatial frequencies, the optically Fourier-transformed light rays that cross the iris opening towards the second half of the 4f system satisfy the following condition:(11)$$\nu_{\rho }=\sqrt{\nu_{x}^{2}+\nu_{y}^{2}}\leq \frac{a}{\lambda f}.$$

Hence, higher spatial frequencies are blocked, potentially reducing fine-detail resolving, i.e., resolution. Nonetheless, the perceived sharpness increases as the zonal and marginal rays are filtered, reducing the effect of lens aberrations, improving the point spread function, and making mid-frequency edges clearer. Consequently, an optimal aperture $a_{\mathrm{opt}}$ provides a cutoff frequency, as shown in Equation (Equation 11), that leverages the effects of lower-frequency contrast and higher-frequency resolution, as studied with tools such as the modulation transfer function. The value of $a_{\mathrm{opt}}$ depends on the imaged object, as well as on the imaging system (through λ and *f*), and is generally hard to assess, making us turn to the use of entropy.

As a metric of *complexity*, the imaging entropy, as defined in Equation (Equation 7), is affected by the number of light rays that arrive at the camera sensor and the structural information that is carried by those rays. In that sense, the optimal aperture $a_{\mathrm{opt}}$ maximizes the imaging entropy as the contributions of both contrast and resolution are balanced according to Equation (Equation 11). A theoretical derivation of this principle, involving the potential analysis of the system’s *étendue*, as presented in ref. [10], is beyond the practical scope intended within this article. Thus, we return to Figure 6 and Figure 7 to inspect the vicinities of the transition region in Section 5.6 and Section 5.7, which will dictate the means for optimization in Section 6.

### 5.6. Above the Transition Region

For apertures bigger than an expected optimal value, the entropy increase can be approximated to a straight line on a logarithmic scale, as seen in Figure 7. That growth behavior is a characteristic feature of the assembly constituted by the imaging system and the imaged object, and it can be used as a reference for determining the transition region for quasi-static scenarios (e.g., slow chemical reactions on a slide as the imaged object). In terms of content, the smaller the aperture, the greater the information about the object, as observed in Figure 5a,b, with the details of the letter “A” becoming sharper. That information, however, is ultimately limited at the transition region, given an imaged object.

### 5.7. Below the Transition Region

At first glance, it might seem inappropriate to assume that the entropy reaches its maximum stable value by looking at Figure 6 and Figure 7. However, if we look at Figure 5c,d, dark patches around the letter “A” start to appear and become more evident as the aperture decreases. These dark patches are imaging artifacts introduced by the imperfections on the object mask, as commented in Figure 3. The smaller the aperture, the lower the contrast and the higher the resolution; thus, the imperfections that were previously neglectable become evident, influencing the entropy values below the transition region.

Note that our analysis assumes all relevant structural information about the object; specifically, the letter “A” on the object mask is fully contained within the imaged area and that no additional structures beyond the letter are of interest. If, instead, one wished to image finer details of the print itself, the imaging system would need to be adjusted, such as by increasing magnification, to appropriately capture the desired structural features. For that reason, the entropy for the letter “A” image (excluding the defects surrounding it) reached its maximum value at the transition region, where one finds an optimal aperture value, thus balancing contrast and resolution.

## 6. Optimization Strategies

Having validated our analytical technique in Section 5, we now offer means to implement it in actual optical systems using algorithms for automated aperture optimization, provided the target optical system can be automatized. In Algorithm 1, we describe an aperture scanning strategy for finding the transition or optimal aperture region, which essentially copies the measurement procedure we adopted to produce Figure 6 and Figure 7. The procedure to calculate the entropy of an image follows the steps in Section 5.2. As the “fitness function” to discriminate the transition region, we used the numerical derivative of the imaging entropy as a function of the aperture, given the behavior of the entropy curves in Figure 6. A natural variant to this approach would be to take the numerical derivative on the log–log scale, given that the entropy curves in Figure 7 have smoother shapes.

As aperture scanning can be time-consuming, we came up with an auxiliary approach to find a smaller upper aperture $a_{\max}$ in Algorithm 2, thus reducing the aperture range in Algorithm 1. We based Algorithm 2 on the vicinal analysis of the transition region in Section 5.6 and Section 5.7.
**Algorithm 1** Optimize Aperture for the Contrast–Resolution Trade-off.**Require:** Minimum aperture $a_{\min}$, maximum aperture $a_{\max}$, aperture step Δa**Ensure:** Optimal aperture region $\left[a_{i},a_{i+1}\right]$1:Initialize empty list A←[]                   ▹ Aperture values2:Initialize empty list H←[]                   ▹ Entropy values3:$a\leftarrow a_{\min}$4:**while** $a\leq a_{\max}$ **do**5:    Capture image $I_{a}$ with aperture *a*6:    Compute entropy $H_{a}\leftarrow \mathrm{Entropy}\left(I_{a}\right)$ (use the procedure described in Section 5.2)7:    Append *a* to A, append $H_{a}$ to H8:    a←a+Δa9:**end while**10:Initialize list of entropy derivatives D←[]11:**for** i=1 to |A|−2 **do**12:    Compute central difference derivative:$$D_{i}\leftarrow \frac{H_{i+1}-H_{i-1}}{a_{i+1}-a_{i-1}}$$13:    Append $D_{i}$ to D14:**end for**15:Identify indices *i* where derivative change is abrupt:$$\mathrm{Find}\;i\;\mathrm{such}\;\mathrm{that}\;|D_{i+1}-D_{i}|\;\mathrm{is}\;\mathrm{maximized}$$16:**return** Aperture region $\left[a_{i},a_{i+1}\right]$ as the optimal aperture range

**Algorithm 2** Estimate of the Upper Aperture ($a_{\max}$) to Reduce Scanning Time.
**Require:** System’s minimum aperture $a_{m}$, system’s maximum aperture $a_{M}$, aperture step Δa, number of points *N*, scaling factor ϵ≥1**Ensure:** Estimated aperture limit $a_{\max}$1:Capture image $I_{m}$ at $a=a_{m}$2:Compute entropy $H_{m}\leftarrow \mathrm{Entropy}\left(I_{m}\right)$ (use the procedure described in Section 5.2)3:Initialize empty lists $H_{M}\leftarrow \left[\right]$, $A_{M}\leftarrow \left[\right]$4:

$a\leftarrow a_{M}$

5:**for** i=1 to *N* **do**6:    Capture image $I_{a}$ at current *a*7:    Compute entropy $H_{a}\leftarrow \mathrm{Entropy}\left(I_{a}\right)$ (use the procedure described in Section 5.2)8:    Append $H_{a}$ to $H_{M}$, append *a* to $A_{M}$9:    a←a−Δa10:
**end for**
11:Take absolute values: $H_{M}\leftarrow \left|H_{M}\right|$12:Normalize aperture values: $A_{M}\leftarrow A_{M}/a_{m}$13:Take logarithms: $\log H_{M}\leftarrow \log\left(H_{M}\right),\;\log A_{M}\leftarrow \log\left(A_{M}\right)$14:Perform linear regression on $\log H_{M}=\alpha \log A_{M}+\beta$15:Compute:$$\log a_{\max}=\frac{\log H_{m}-\beta }{\alpha }$$16:

$a_{\max}\leftarrow \exp\left(\log a_{\max}\right)$

17:**return** $\epsilon \cdot a_{\max}$


## 7. Conclusions

In this paper, we have described an analytical technique for measuring the statistical entropy of an imaged object as a function of the imaging system’s aperture, enabling the identification of an aperture value that optimizes resolution and contrast to yield a detailed and well-illuminated image. Our approach draws inspiration from the premises of the *H*-theorem, which explains how certain nonequilibrium systems evolve towards equilibrium while employing the spatial filtering properties of Fourier optics.

As a case study of our technique, we have built a 4f system to image an object mask with imperfections, as discussed in Section 4. We have imaged that target at various aperture values, determining the entropy of each image, as presented in Section 5. We found a transition region where the entropy as a function of the aperture changes its behavior. Above that region, entropy increases as the aperture decreases. Below that region, details from the imperfections become evident, affecting the entropy curve. Therefore, the transition region is where the optimal aperture is, with the intended details of an object in evidence.

On the one hand, an important assumption for our technique is that the desirable information about an imaged object is fitted into the imaged area. Smaller structures within that area might not be as resolved as the structures that fill the characteristic dimensions under imaging. Furthermore, it also assumes a static or quasi-static target at the object plane.

On the other hand, our analytical technique has the advantage of not relying on heavy mathematical modeling and physical treatment, as many other approaches usually do. In fact, it relies mainly on the content that comes with the definition of entropy, making it a very robust method for optical imaging enhancement. In terms of applications, while this technique does require aperture scanning, this requirement enables automated optimization, as presented in Section 6, making it particularly attractive for improving optical scanners, confocal microscopes, and other Fourier optics-based systems.

## Figures and Tables

**Figure 1 entropy-27-00730-f001:**
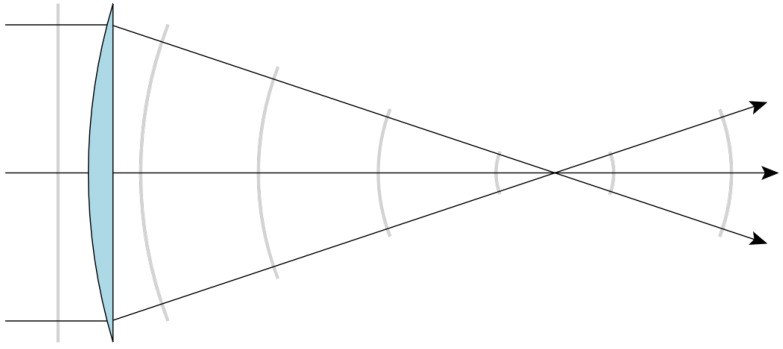
The curved surface of a thin plano-convex lens maps an incident plane wavefront into a spherical wavefront. By placing a point source on the focal point and flipping the curved surface towards it, the curved surface now maps the incident spherical wavefront into a plane wavefront. Source: https://commons.wikimedia.org/wiki/File:Spherical_wave_lens2.svg (accessed on 24 June 2025)—CC BY-SA 3.0.

**Figure 2 entropy-27-00730-f002:**
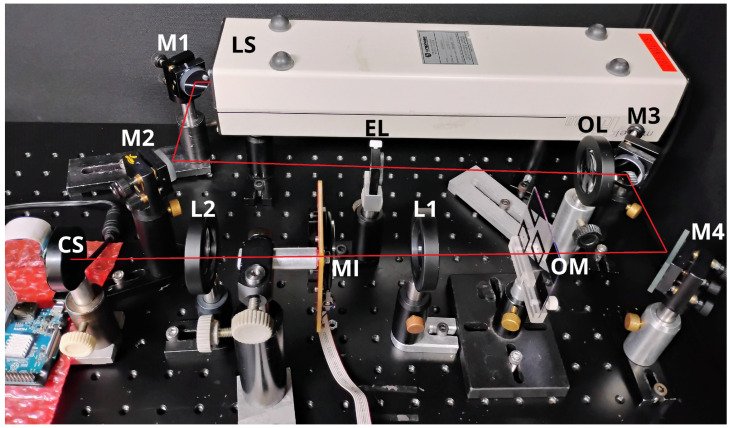
Optical system assembled for our experiment. The red-line segments indicate the path of the light throughout the system, which was produced by a 632.8-nm helium–neon laser source (LS) and whose height and slope of the laser beam was precisely controlled by a set of four aluminum mirrors (M1, M2, M3, and M4). Between M2 and M3, a Keplerian telescope was set up, using an eyepiece lens (EL) of f=20 mm and an objective lens (OL) of f=120 mm, producing a collimated beam of approximately 1 cm in diameter at the back of OL. That collimated beam was guided by M3 and M4 to a 4f system using two biconvex lenses (L1 and L2) of f=75 mm. At the object plane, the object mask (OM) shown in Figure 3 was positioned, imprinting a letter “A” on the beam as a shadow. At the Fourier plane (where L1 and L2’s foci coincide), a remotely operable motorized iris (MI) was placed for aperture control. At the image plane, a camera sensor (CS) was settled: a lensless 5-megapixel camera module connected to a *Raspberry* board (partly visible at the lower left corner).

**Figure 3 entropy-27-00730-f003:**
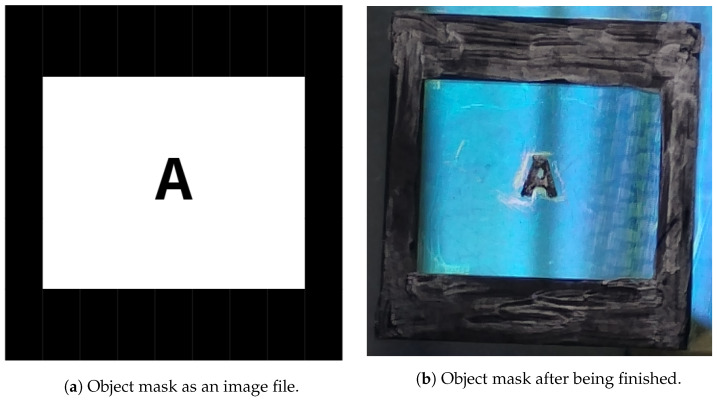
A letter “A” absorptive mask used as the object mask (OM) in the optical system of Figure 2. The design shown in (**a**) was initially printed on a glossy paper sheet using a high-quality printer. Then, the printed layout was cut from the sheet and placed over a piece of anti-reflective coated glass, secured by tape at the edges. Next, the paper–glass assembly was subjected to a heated press, commonly used for small-scale printed-circuit-board fabrication, to transfer the printed layout to the glass surface. However, due to the poor quality of the glossy paper, a thin layer of that material remained on the glass. That residue was manually removed using fingertip friction, which damaged parts of the print in the process. Around the letter “A”, a small file was used to avoid damaging the character itself, which in turn left scratches on the surrounding surface. Finally, a black marker pen was used to fill in the damaged areas on the print. The object mask is seen in (**b**) reflecting the blue spectrum of the ceiling light, highlighting artifacts introduced during the making process, which will influence the results in Section 5.

**Figure 4 entropy-27-00730-f004:**
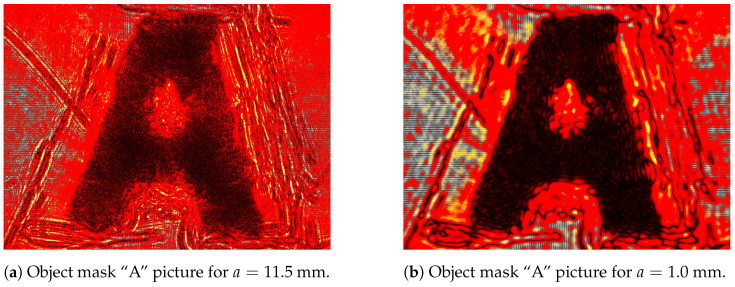
Object mask “A” for maximum (a) and minimum (b) apertures. It is visually clear that both contrast and resolution are enhanced by decreasing the aperture. The goal of our analytical technique is to find the best compromise between contrast and resolution solely using the entropy.

**Figure 5 entropy-27-00730-f005:**
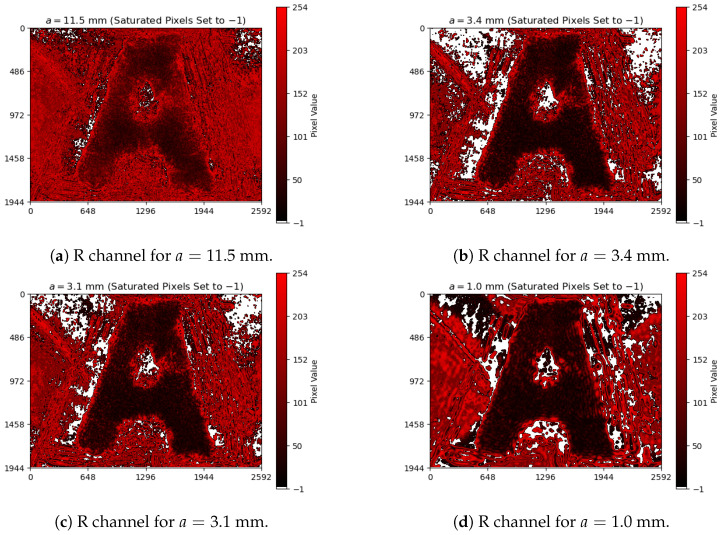
Selected colormap images from Appendix A to analyze the diagrams in Figure 6 and Figure 7: the highest-aperture (**a**) and lowest-aperture (**b**) images above the transition region, and the highest-aperture (**c**) and lowest-aperture (**d**) images below the transition region. Saturated and high-variance pixels are not used for entropy calculations in Section 5.2.

**Figure 6 entropy-27-00730-f006:**
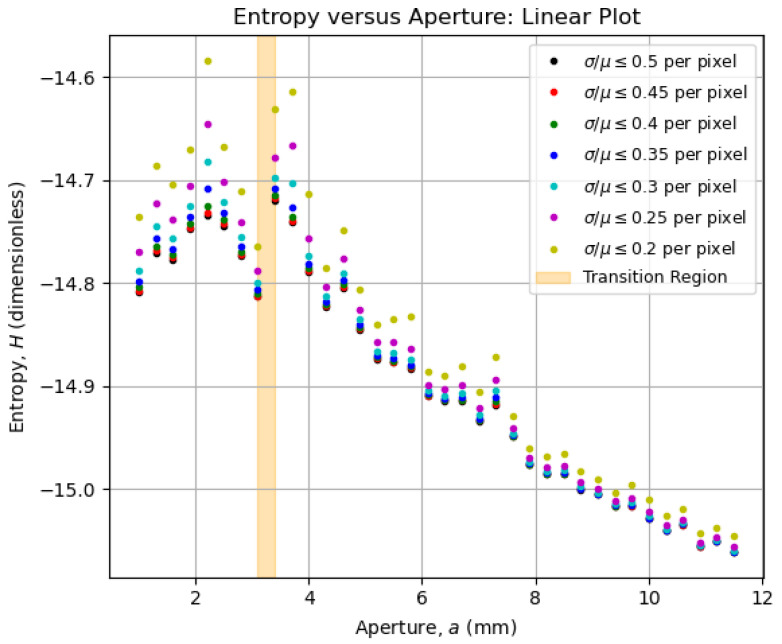
Behavior of the entropy as a function of the aperture for the images of the object mask in Figure 3, which were acquired with the imaging system in Figure 2. The curves retain their overall shapes for coefficient-of-variation thresholds up to 0.2, indicating the limit of statistical reliability. There is a clear change in the behavior of the curves between a=3.4 mm and a=3.1 mm (orange-shaded area), marking the region of optimal aperture for the best trade-off between resolution and contrast.

**Figure 7 entropy-27-00730-f007:**
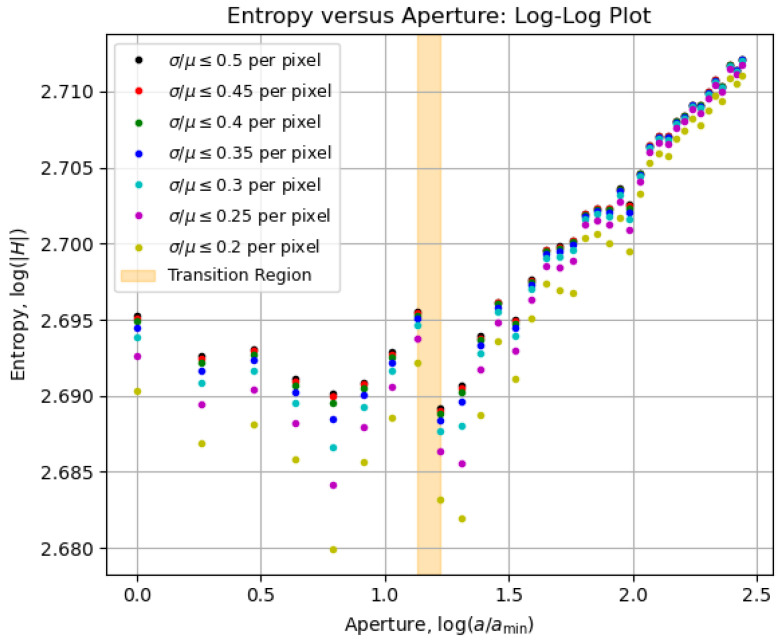
Log–log-scale version of the diagram in Figure 6, in which the smoothening effect of the logarithm function on the curves makes their behavior more evident. Note how little the data vary for apertures below the transition region, pointing at expected stability below the optimal aperture.

**Figure 8 entropy-27-00730-f008:**
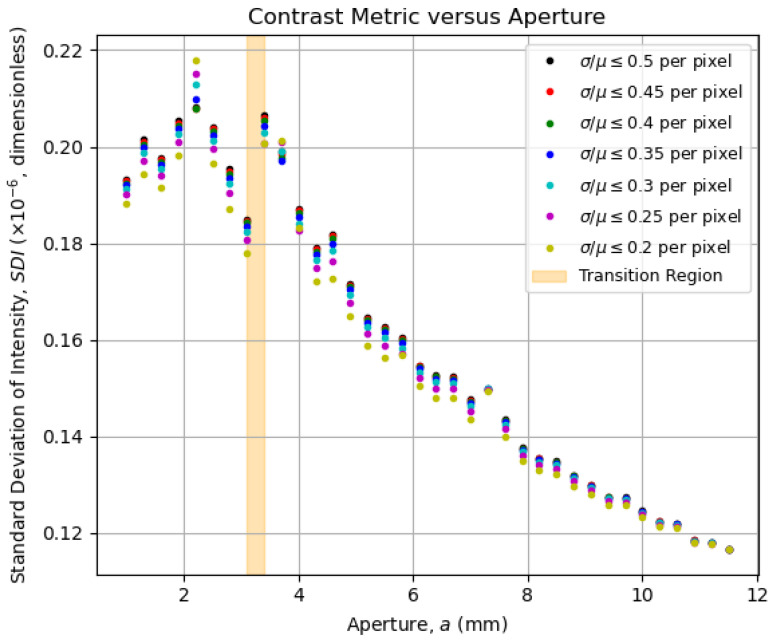
Contrast metric defined in Equation (Equation 9) as a function of the aperture for fixed coefficient-of-variation thresholds. Notice how the contrast metric increases as the aperture decreases, changing its behavior at the transition region. For lower apertures, the contrast metric does not vary significantly.

**Figure 9 entropy-27-00730-f009:**
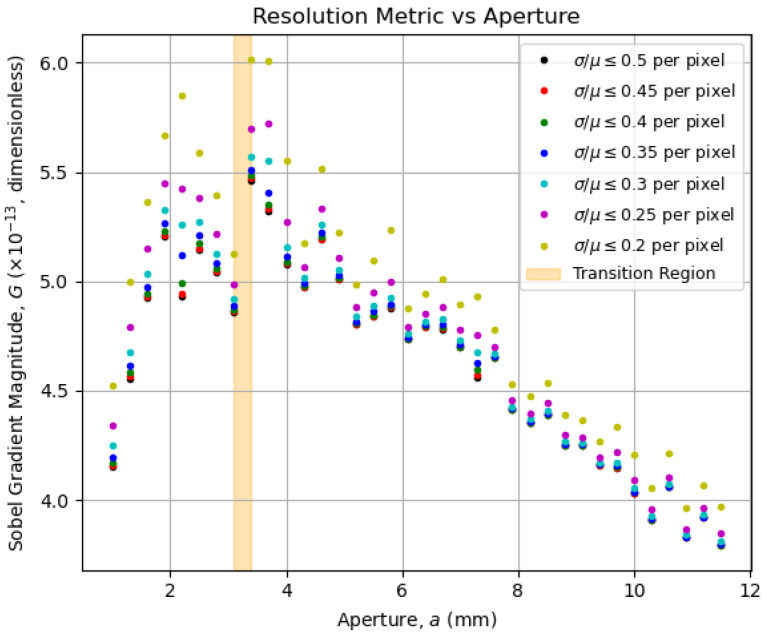
Resolution metric defined in Equation (Equation 10) as a function of the aperture for fixed coefficient-of-variation thresholds. As the contrast metric, the resolution metric increases as the aperture decreases, also changing its behavior at the transition region. However, it varies more significantly at lower apertures, as the dark patches due to the object mask imperfections become more resolved, as seen in Figure 5c,d, causing the resolution metric to increase again.

## Data Availability

The minimum dataset was provided with the submission. The full dataset is available upon request due to file size limitations.

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
