# Peer review of "Entropy-Inspired Aperture Optimization in Fourier Optics"

_entropy, 2025, doi:10.3390/e27070730_

Round 1
Reviewer 1 Report
Comments and Suggestions for Authors
This manuscript proposes an aperture optimization method for Fourier optical systems based on the variation of image entropy, aiming to identify an optimal trade-off between image resolution and contrast through entropy maximization. The authors use a 4f optical system as the experimental platform, scanning the aperture at the Fourier plane and analyzing the pixel intensity distribution entropy of captured images to determine the aperture range associated with the highest image information content and quality. The problem addressed in this study is of considerable importance and practical relevance. However, In terms of technical novelty, the proposed method is basic, involving only a straightforward entropy calculation applied to standard image acquisition. It does not involve any new theoretical derivation or algorithmic framework and primarily constitutes a low-threshold empirical exploration. So I do not believe this manuscript meets the standards for publication as a formal research paper. Specific comments follow:
- Although the study employs a physically motivated analogy to the Boltzmann H-theorem and provides experimental validation with a handcrafted object-mask, the analogy lacks formal rigor. No well-defined or theoretically justified mapping exists between thermodynamic entropy and optical image formation, which risks leading to misleading interpretations.
- Moreover, the manuscript fails to establish either an analytical or empirical relationship between aperture size and image quality metrics such as contrast or resolution. Although a large number of images were acquired across different aperture values, no explicit functional dependence or optimization criterion is presented.
- The procedures for masking saturated pixels and handling background noise are not clearly described. It remains unclear whether such preprocessing steps are necessary in practical applications, and no standard criteria for preprocessing are defined. These steps could significantly affect the entropy calculations, potentially undermining the reliability of the final conclusions.
- The proposed method identifies the optimal aperture based solely on image entropy, without comparing it against existing image sharpness evaluation metrics such as variance, edge intensity, or established entropy-based measures. As a result, it is difficult to assess the relative performance or advantages of the proposed approach.
- The core claim—that there exists an aperture function range (that is the transition region proposed in the manuscript - the optimal aperture) within which the image simultaneously achieves optimal resolution, contrast, and information content—is presented without sufficient justification. The experimental validation is based on a single handcrafted target with uncontrolled imperfections, severely limiting the generalizability of the results. The images provided do not clearly demonstrate improvements in contrast or resolution, nor do they offer quantitative evidence or image quality metrics to support the claims.Moreover, the imaging quality in the study is truly unacceptable.
- The manuscript lacks a discussion on the physical meaning of the "optimal aperture" and its dependence on system parameters. Does this "optimal value" depend on the wavelength of the light source, the object distance, the magnification, and the size of the obstruction? Does it have the ability of system generalization? This is crucial to the practicability of the method.
- The improvement of resolution should be tested with standard samples such as resolution plates instead of directly observing the imaging of the object mask 'A' .
Author Response
This manuscript proposes an aperture optimization method for Fourier optical systems based on the variation of image entropy, aiming to identify an optimal trade-off between image resolution and contrast through entropy maximization. The authors use a 4f optical system as the experimental platform, scanning the aperture at the Fourier plane and analyzing the pixel intensity distribution entropy of captured images to determine the aperture range associated with the highest image information content and quality. The problem addressed in this study is of considerable importance and practical relevance. However, In terms of technical novelty, the proposed method is basic, involving only a straightforward entropy calculation applied to standard image acquisition. It does not involve any new theoretical derivation or algorithmic framework and primarily constitutes a low-threshold empirical exploration. So I do not believe this manuscript meets the standards for publication as a formal research paper. Specific comments follow:
Dear reviewer, we greatly appreciate your constructive criticisms, which pointed out important shortcomings in our arguments and have helped improve our manuscript. While it is true that our experimental design is simple, we believe that this simplicity is a key strength of our proposed method. The revisions now clarify the relationship between entropy and aperture, providing both a validation of our technique and a framework for algorithmic optimization. We aim to demonstrate this by addressing each of your specific comments below.
-
Although the study employs a physically motivated analogy to the Boltzmann H-theorem and provides experimental validation with a handcrafted object-mask, the analogy lacks formal rigor. No well-defined or theoretically justified mapping exists between thermodynamic entropy and optical image formation, which risks leading to misleading interpretations.
The formal theory of imaging entropy is provided in Ref. 8 and summarized in our manuscript by Eq. (6). However, we acknowledge that we did not explicitly present the relationship between the measured imaging entropy and the aperture in the Fourier plane. To address this, we have expanded Section 3 to include the explicit equations. Additionally, we discussed the H-theorem in Section 2 as a motivation for introducing our ideas; however, we did not interpret our results beyond their numerical values and the metrics employed.
-
Moreover, the manuscript fails to establish either an analytical or empirical relationship between aperture size and image quality metrics such as contrast or resolution. Although a large number of images were acquired across different aperture values, no explicit functional dependence or optimization criterion is presented.
To address this issue, we have renamed Section 5 (originally titled “Results and Discussion”) to “Validating the Technique”, where we aim to demonstrate the functional dependence of entropy on aperture in terms of contrast and resolution (see our response to Comment 5). Building on Section 5, we have added a new section (Section 6, “Optimization Strategies”) to present optimization criteria based on our proposed procedures.
-
The procedures for masking saturated pixels and handling background noise are not clearly described. It remains unclear whether such preprocessing steps are necessary in practical applications, and no standard criteria for preprocessing are defined. These steps could significantly affect the entropy calculations, potentially undermining the reliability of the final conclusions.
The saturated-pixel masking procedure is described in Subsection 5.1, while the treatment of background noise during entropy calculation is detailed in Subsection 5.2.
-
The proposed method identifies the optimal aperture based solely on image entropy, without comparing it against existing image sharpness evaluation metrics such as variance, edge intensity, or established entropy-based measures. As a result, it is difficult to assess the relative performance or advantages of the proposed approach.
Expanding the analysis of the entropy curves in Subsection 5.3, we have added a new subsection (Subsec. 5.4, “Contrast and Resolution Metrics”) to examine how contrast (measured by the standard deviation of intensity) and resolution (measured by the average squared gradient magnitude) vary with aperture, and their behavior relates to the entropy.
-
The core claim—that there exists an aperture function range (that is the transition region proposed in the manuscript - the optimal aperture) within which the image simultaneously achieves optimal resolution, contrast, and information content—is presented without sufficient justification. The experimental validation is based on a single handcrafted target with uncontrolled imperfections, severely limiting the generalizability of the results. The images provided do not clearly demonstrate improvements in contrast or resolution, nor do they offer quantitative evidence or image quality metrics to support the claims.Moreover, the imaging quality in the study is truly unacceptable.
We hope to address these issues through the newly added sections, as discussed in our responses to the other specific comments. Additionally, the images presented in the previous version of the manuscript were colormap plots of the R-channel data, not the actual images captured by the camera. In the revised version, we now include two actual images of the object-mask “A” (one taken at the maximum aperture and one at the minimum aperture) in the newly added Figure 4 at the beginning of Section 5, showing how different they are from their corresponding colormap plots.
-
The manuscript lacks a discussion on the physical meaning of the "optimal aperture" and its dependence on system parameters. Does this "optimal value" depend on the wavelength of the light source, the object distance, the magnification, and the size of the obstruction? Does it have the ability of system generalization? This is crucial to the practicability of the method.
We have added a new subsection (Subsec. 5.5) discussing the physical meaning of the optimal aperture—or transition region—in terms of the 4f system’s operation and the entropy metrics. The behavior of the entropy curves above and below this transition region, previously discussed in our manuscript, is now presented in Subsecs. 5.6 and 5.7.
-
The improvement of resolution should be tested with standard samples such as resolution plates instead of directly observing the imaging of the object mask 'A' .
Unfortunately, we did not have resolution plates at our disposal during the experiments presented in this manuscript. However, given the practical focus of our method—specifically in relation to the quantities we aimed to measure—we believe we were able to adequately address this limitation, as discussed in our responses to the previous specific comments.
Reviewer 2 Report
Comments and Suggestions for Authors
1 Overall, the introduction is well-founded and includes all relevant references. The experimental design is appropriate for the problem.The choice of a simple static target (a letter “A” mask with intentional imperfections) is sensible for testing the method under realistic conditions. While the study focuses on a single object, the design is adequate to demonstrate the concept of entropy-guided aperture optimization. The results are presented in a clear and systematic manner. The manuscript clearly separates analysis above and below the transition region, arguing convincingly that within this region lies the best compromise between resolution and contrast. In summary, the results section is clearly written and the entropy-versus-aperture analysis is presented with appropriate rigor. One suggestion for further clarity might be to quantify the “abrupt change” defining the transition region (for example, via a numerical criterion or error bars on the entropy curves), but qualitatively the evidence is clear.The conclusions do not overreach beyond what the data show. In sum, the experimental results presented (entropy curves and example images) fully back up the paper’s conclusions.
2 All figures are clear, well-labeled, and support the text. One minor recommendation is to include error bars or uncertainty ranges in Figures 5 and 6 for completeness, given that multiple images were averaged – however, the current presentation is still clear.
3 Another minor suggestion would be for the Appendix. If all redundant colorbars and legends were removed (after all the aperture is indicated in the titles) all 36 figures could be fit into a 6x6 panels composite figure, where the reader could explore the differences at a glance, because scrolling all the figures in a 2-column layout is not optimal for the reader.
4 Finally, I noted only a few very minor issues in English (e.g. the term “exposition time” should likely be “exposure time”), but these do not affect readability.
Author Response
1 Overall, the introduction is well-founded and includes all relevant references. The experimental design is appropriate for the problem.The choice of a simple static target (a letter “A” mask with intentional imperfections) is sensible for testing the method under realistic conditions. While the study focuses on a single object, the design is adequate to demonstrate the concept of entropy-guided aperture optimization. The results are presented in a clear and systematic manner. The manuscript clearly separates analysis above and below the transition region, arguing convincingly that within this region lies the best compromise between resolution and contrast. In summary, the results section is clearly written and the entropy-versus-aperture analysis is presented with appropriate rigor. One suggestion for further clarity might be to quantify the “abrupt change” defining the transition region (for example, via a numerical criterion or error bars on the entropy curves), but qualitatively the evidence is clear.The conclusions do not overreach beyond what the data show. In sum, the experimental results presented (entropy curves and example images) fully back up the paper’s conclusions.
Dear reviewer, we appreciate your comments, which have contributed to improving our manuscript. The text has been significantly expanded to address the points raised by another reviewer, while the original content has been preserved with only minor modifications. Section 5 (originally “Results and Discussion”) has been renamed Validating the Technique and now includes an expanded analysis of the results. We have also added a new section (Section 6, “Optimization Strategies”) to address the quantification of the observed 'abrupt change' for practical implementation. Your specific comments are addressed in our responses below.
2 All figures are clear, well-labeled, and support the text. One minor recommendation is to include error bars or uncertainty ranges in Figures 5 and 6 for completeness, given that multiple images were averaged – however, the current presentation is still clear.
To assess the statistical stability of our data—particularly since multiple images were averaged—we chose to present the entropy curves using the coefficient of variation (CV) instead of traditional error bars. Although this approach is not commonly used in the physical sciences, it serves as a powerful visualization tool. For this reason, we have chosen to retain the current format of the diagrams. However, if deemed necessary, we are open to including a subsection explaining the key features and rationale behind the use of CV.
3 Another minor suggestion would be for the Appendix. If all redundant colorbars and legends were removed (after all the aperture is indicated in the titles) all 36 figures could be fit into a 6x6 panels composite figure, where the reader could explore the differences at a glance, because scrolling all the figures in a 2-column layout is not optimal for the reader.
The colormap plots in App. A are now presented as a 4x9 panel, utilizing the available page space more effectively. Additionally, we have included a second 4x9 panel on the following appendix page, showing the corresponding standard deviation plots.
4 Finally, I noted only a few very minor issues in English (e.g. the term “exposition time” should likely be “exposure time”), but these do not affect readability.
We have reviewed the previous text to address minor issues. However, as the manuscript has been significantly expanded in this new version, a further review may still be necessary. We apologize in advance for any errors—minor or otherwise—that may remain.
Round 2
Reviewer 1 Report
Comments and Suggestions for Authors
The authors have revised the paper according to the comments one by one. This version is ready for publication now.
Reviewer 2 Report
Comments and Suggestions for Authors
The authors have addressed all the issues raised in the review, and I have no further comments.
Comments on the Quality of English LanguageThe authors have corrected some minor typos, and the overall clarity and use of language have been significantly improved, resulting in a more polished and professional manuscript.